# Establishing an IPM System for Tarnished Plant Bug (Hemiptera: Miridae) in North Carolina

**DOI:** 10.3390/insects16020164

**Published:** 2025-02-05

**Authors:** Dominic Reisig, Anders Huseth

**Affiliations:** 1Department of Entomology and Plant Pathology, NC State University, 207 Research Station Rd., Plymouth, NC 27962, USA; 2Department of Entomology and Plant Pathology, the North Carolina Plant Sciences Institute, NC State University, 840 Oval Dr., Raleigh, NC 27606, USA; ashuseth@ncsu.edu

**Keywords:** landscape, sampling, insecticide recommendation, ThryvOn

## Abstract

The tarnished plant bug was once a minor pest in North Carolina cotton fields, but has become a major problem in recent years. This insect has long been a persistent pest in other cotton-growing regions, like the Midsouth. Extension entomologists have worked to understand the pest’s behavior and improve management strategies in North Carolina. In this region, tarnished plant bugs are more common in cotton fields near crops like wheat, soybeans, and corn, and in fields with many edges. Extension entomologists tested how best to estimate tarnished plant bug populations and found practical methods for different stages of cotton growth. They also confirmed that pest management practices from the Midsouth work well in North Carolina and developed an insecticide plan that balances pest control with protecting beneficial insects. Ongoing studies are evaluating new tools, like a new type of Bt cotton called ThryvOn, to manage the pest more effectively. These efforts aim to provide cotton growers with sustainable pest management strategies that protect their crops and reduce the need for excessive insecticide use.

## 1. Introduction

The tarnished plant bug, *Lygus lineolaris* Palisot de Beauvois, is a piercing–sucking insect in the family Miridae. In cotton, *Gossypium hirsutum* L., *Lygus* spp. can cause significant damage, including plant height and weight reductions, terminal abortion, square (pre-floral bud) abscission, malformed bolls, boll abscission, stained lint, and damaged seed. These effects can delay maturity and lead to yield losses [1,2,3,4,5]. Tarnished plant bug has been a significant pest of cotton in the Midsouth U.S. (Arkansas, Louisiana, southeastern Missouri, Mississippi, and eastern Tennessee) since at least the 1970s [3,6], and became increasingly problematic in the region starting in 1995 [7]. By 2008, it had become the most expensive insect pest to manage on a per-acre basis across the Midsouth [8].

The stink bug complex, also piercing–sucking pests, grew in prominence during the same period. By 2008, stink bugs were the second-most costly insect pests to manage on a per-acre basis [8]. The increased economic significance of the piercing–sucking pest complex has been attributed to several significant changes to the cotton production system. The successful eradication of boll weevil, *Anthonomus grandis* Boheman (Coleoptera: Curculionidae), and the widespread planting of Bt cotton reduced the use of foliar insecticides, likely creating an ecological niche for former secondary pests like tarnished plant bug and stink bugs [9,10,11]. Additional factors, such as the shift to relatively short-lived neonicotinoid seed treatments in place of longer-lasting in-furrow insecticides, reduced tillage practices associated with glyphosate-tolerant crop varieties, and the emergence of insecticide-resistant populations may have further contributed to this trend [12]. Similarly, in China, the adoption of Bt cotton has been strongly correlated with reduced insecticide use, leading to an increase in mirid pests across multiple crops [13,14]. These observations suggest that reduced insecticide environments can favor the proliferation of secondary pests, though other contributing factors are likely.

In the southeastern U.S. cotton production region (Alabama, Georgia, North Carolina, South Carolina, and southeastern Virginia), the situation evolved differently. After boll weevil eradication and the introduction of Bt cotton, tarnished plant bug was not a major pest of cotton in this region, but stink bugs were among the most costly pests to manage [8]. For instance, in North Carolina in 2009, 40% of the cotton acreage was sprayed for stink bugs, while only 1% was treated for tarnished plant bug [15]. This changed dramatically in 2010, when tarnished plant bug populations in cotton began to increase across the southeastern U.S., especially in southeastern Virginia and northeastern North Carolina (Figure 1). This increase occurred despite continued use of broad-spectrum insecticides targeting bollworm, *Helicoverpa zea* Boddie (Lepidoptera: Noctuidae), and stink bugs, which should have indirectly suppressed tarnished plant bug populations [16,17,18,19]. The introduction of pyramided Bt toxin varieties in 2003, which required fewer bollworm-targeted applications, does not fully explain this rise, as single-toxin Bt varieties were still commonly planted, and insecticide use targeting bollworm remained high through 2012 [20]. Thus, the simple explanation of a reduced insecticide environment is insufficient to account for the increased prevalence of tarnished plant bug in North Carolina.

Despite the unclear causes, tarnished plant bug had become a significant pest in North Carolina by 2017, with 75% of the state’s cotton acreage treated for this insect [21]. This review describes the establishment and impact of an integrated pest management (IPM) plan for tarnished plant bug in North Carolina cotton. These efforts have stabilized insecticide use targeting this pest, preserved beneficial insects, and likely prevented secondary pest outbreaks. Furthermore, this study highlights how an IPM system developed for one region (the midsouth U.S. cotton production region; largely outlined in [22,23,24,25]) can be effectively adapted to another (the southeast U.S. cotton production region). In this review, we document research conducted in the southeast U.S. cotton production region on landscape effects, sampling, thresholds, and insecticides, which led to an effort to evaluate the effects of these in a study to evaluate IPM for tarnished plant bug in North Carolina and Virginia. We end the review touching on a new management tactic (Bt cotton targeted for tarnished plant bug) and the need to incorporate this into the existing IPM system.

## 2. Landscape Effects

Plant host quality from both a feeding and reproductive perspective is important to the seasonal population dynamics of tarnished plant bug. For example, while cotton serves as a feeding and reproductive host for *Lygus* spp. [26,27,28], both non-cultivated and cultivated hosts are important to several generations of *Lygus* spp. prior to feeding on cotton [29,30,31]. Furthermore, while many of these are superior hosts compared to cotton [26,27,28,32,33], when alternate host quality declines, *Lygus* spp. adults disperse into cotton [29,33,34,35,36,37].

In North Carolina and Virginia, prior to 2010, tarnished plant bug was usually an issue in cotton when there were known alternate hosts in the environment, such as cultivated potato, *Solanum tuberosum* L., weedy field borders with crucifers, or cultivated clary sage, *Salvia sclarea* L. However, as mentioned previously, tarnished plant bug populations began to increase in cotton across the southeastern U.S. after this time. Changes in management practices did not seem to be associated with this population increase, leading some to hypothesize that the answer was more complex.

A landscape analysis in Virginia found that cotton fields near double-cropped wheat and soybean, *Glycine max* (L.) Merr., fields, corn, *Zea mays* L., fields, disturbed forest, or in areas of higher agricultural intensity tended to have higher populations of tarnished plant bug [38]. Additionally, cotton fields with higher annual minimum temperatures tended to have higher populations of tarnished plant bug. Another study across the southeastern U.S. found more tarnished plant bugs in cotton fields near double-cropped wheat, *Triticum aestivum* L., and soybean fields, near other cotton fields, and in areas of higher agricultural intensity [39]. Of those factors, areas of higher agricultural intensity tended to have more influence on tarnished plant bug abundance in cotton than proximity to double-cropped wheat and soybean fields or cotton fields. A survey of 69 wheat fields in an intensive cotton producing region of North Carolina confirmed that the crop could be a major source patch for tarnished plant bug in the spring [40]. Finally, a North Carolina study found higher numbers of nymphs in field edges and in fields with more accumulated degree days [41]. Adult numbers tended to be higher in fields with more fragmentation in the landscape, which results in more field edges and weedy non-crop habitat.

While only correlative, these findings suggest that changes in landscape composition and increasing minimum temperatures may have played roles in the emergence of significant tarnished plant bug populations in cotton in the region. However, other unexplored factors may also have contributed to the regional emergence of this pest. This highlights the need for more holistic integrated pest management strategies that consider environmental factors from the field to landscape scale.

## 3. Sampling

*Lygus* spp. are distributed in clumped, nonrandom patterns in cotton fields [42], making sampling challenging compared to more uniformly distributed pests. Furthermore, tarnished plant bug nymphs are distributed with more variability relative to the edge of a cotton field than adults [41]. To more accurately assess population distributions, standardized sampling plans were evaluated across the southeastern U.S. From this study, the minimum required sample for estimating tarnished plant bug populations at a 0.25 precision level was determined to be 12 sampling locations throughout the field with 25 sweeps per location (pre-bloom) and 10 locations of a single drop cloth sample in a field [39]. Across the region, there was significant variation in which fields reached the economic threshold. This highlights the importance of sampling each field independently, ensuring management decisions for tarnished plant bug are based on field-level population assessments rather than regional averages. This study also showed that the sampling effort for accurate threshold determination may exceed typical amounts used by independent crop consultants in the region, which may lead to inaccurate sampling results.

## 4. Thresholds

Economic thresholds developed in the Midsouth took advantage of the strengths of different sampling methods to achieve precise assessments depending on the phenological stage of the crop. For example, both the sweep net and drop cloth are precise sampling methods, but sweep net samples are biased toward adult capture, while drop cloth samples are biased toward nymph capture [10]. Midsouth studies conducted pre-bloom confirmed the validity of a previously established threshold of eight tarnished plant bugs per 100 sweeps combined with 80% square retention [12]. This takes advantage of the fact that adults are typically moving into pre-bloom cotton fields to feed on squares, their preferred cotton tissue [43]. Furthermore, the addition of square retention to the threshold helped minimize economic costs [12] by adding the dimension of plant compensation to tarnished plant bug square injury. In addition, studies in blooming cotton confirmed the economic injury level between 1.6 and 2.6 tarnished plant bugs per drop cloth sample [7].

In 2016 and 2017, Midsouth thresholds were validated in both southeastern Virginia and northeastern North Carolina [5]. In this study, post-bloom insecticide applications were more important to preserve yield in earlier planted cotton, while pre-bloom and early-bloom insecticide applications were more effective for yield in later-planted cotton. Furthermore, economic returns for managing tarnished plant bugs were higher in earlier-planted cotton, underscoring the importance of early maturity in the upper southeastern U.S. This area is the northernmost U.S. cotton production region in the eastern Cotton Belt. Due to the region’s short growing season, squares produced prior to bloom are typically more important contributors to yield in late-planted cotton, while earlier planted cotton can compensate for pre-bloom square loss during bloom.

## 5. Insecticides

Insecticide management for tarnished plant bug is an ongoing challenge due to changing pesticide options with different frequency of use and residual activity. In the 1990s, they evolved resistance to cyclodienes, pyrethroids, and organophosphates in the Midsouth, likely selected by insecticide applications targeted for tobacco budworm, *Chloridea virescens* Fabricius (Lepidoptera: Noctuidae) [44]. Entomologists at the time recommended that growers voluntarily restrict the use of pyrethroids during the growing season, thereby preserving their efficacy later in the season (ibid.). Despite this recommendation, pyrethroid and organophosphate resistance evolved widely and persisted in Midsouth tarnished plant bug populations [45,46].

Foliar insecticide use patterns in cotton were similar between the Midsouth and the Southeast during the 1990s and 2000s, with reliance on pyrethroids, organophosphates, and the introduction of neonicotinoids. Pyrethroids were the primary class of insecticides used to target tarnished plant bug in the Southeast and were highly effective during 2010 [16], when the pest began expanding in the region. During 2016, independent crop consultants in northeastern North Carolina began reporting poor control using pyrethroids for tarnished plant bug. A replicated small-plot trial in that region demonstrated that total plant bug numbers were statistically the same in plots treated with a pyrethroid compared to untreated plots [47]. However, tarnished plant bug numbers were well controlled in plots where the pyrethroid had been combined with an organophosphate. Later studies confirmed widespread resistance to pyrethroids, and elevated resistance levels to organophosphates and neonicotinoids in a few populations [48].

Midsouth entomologists realized that insecticides would not solve the tarnished plant bug issue, especially given the pest’s propensity to evolve resistance. Furthermore, they noted only “marginal benefits for tarnished plant bug pest management” for the newly available insecticide, sulfoxaflor [25]. As a result, they created a management system optimized by focusing on the application technology (proper nozzle selection and application method), application intervals, insecticide rotation, and application termination timing [24]. In addition, a newly available insecticide, novaluron, showed excellent efficacy in their trials, especially when applied pre-bloom [22]. Based on these results, they adjusted their IPM system to include sequential application of strategically organized insecticides (Figure 2) to manage tarnished plant bug in Midsouth cotton [23].

Previous insecticide screening trials in North Carolina had not identified many classes of insecticides that were consistently effective for tarnished plant bug except for pyrethroids and organophosphates [16,47]. These data, combined with data from Mississippi State, were enough to receive an emergency exemption to use sulfoxaflor in North Carolina cotton during 2018. This insecticide received full registration in 2020.

Subsequent North Carolina insecticide screening data demonstrated that novaluron, widely recommended in the Midsouth, could extend the efficacy of a widely used organophosphate, acephate [49,50]. This is likely due to its activity as an insect growth regulator, functioning to suppress egg eclosion and nymph development [51]. However, the efficacy of pre-bloom applications of novaluron in North Carolina has been inconsistent, possibly due to differences in pest dynamics between the regions. In the Southeast, the timing of tarnished plant bug adult movement into cotton fields and subsequent nymph development vary by location and year [38]. In contrast, tarnished plant bugs are more consistent and predictable pests in the Midsouth, with relatively discrete periods of widespread movement. Therefore, early applications in the Midsouth could help suppress the more regular development of tarnished plant bug in cotton.

In North Carolina during 2016, the same initial year that tarnished plant bug populations were difficult to manage with pyrethroids, bollworm resistance to Bt cotton led to widespread foliar insecticide applications for this pest [20]. In cotton, most bollworm infestations occur after the initiation of bloom [52]. Furthermore, broad-spectrum insecticides effective for mirid pests can flare bollworms, and related species, by reducing natural enemies that suppress their populations [53,54]. Therefore, insecticides that are not as impactful on natural enemies, but still effective for tarnished plant bug, would be helpful in a rotation to avoid flaring bollworm.

To address near-term management challenges, North Carolina extension recommendations for insecticides were inspired by the Midsouth recommendations (Figure 2) and were created based on the expected number of applications in a growing season. In North Carolina, neonicotinoids are the recommended choice of insecticide during the early season (Figure 3). Efficacy for this class of insecticide is initially high in the Southeast, but decreases as the season progresses, perhaps due to increased activity of detoxification enzymes [48] from insecticide exposure across the season. Moreover, while this class has some impact on natural enemies, effects are not as great as other broad-spectrum classes [55]. As a result, neonicotinoids may allow natural enemy populations to recover quicker than other broad-spectrum insecticides.

During squaring and bloom, sulfoxaflor is recommended in North Carolina. At the highest labeled rate, this insecticide is effective for tarnished plant bug and has negligible impacts on natural enemies [56,57]. Similar to the Midsouth, organophosphates and pyrethroids combined with organophosphates are only recommended in North Carolina for the mid- to late-season, after the threat of bollworm has diminished. These insecticide classes are also the only recommended insecticides for stink bugs, which prefer to feed on medium- to large-sized bolls [9,58,59]. Bolls of this size are not present until at least the second and third week of bloom. Therefore, the mid- to late-season insecticides recommended for tarnished plant bug are also effective for stink bugs when they are present.

During the mid- to late-season period, North Carolina cotton growers who typically apply insecticides multiple times per season for tarnished plant bugs are recommended to tank mix novaluron with a broad-spectrum insecticide (Figure 3B,C). The approach combines a fast-acting broad-spectrum insecticide that has activity on adults and nymphs, with novaluron, which works more slowly to inhibit nymph development over a longer duration. In contrast, growers who typically spray once or twice for tarnished plant bug (Figure 3A) are unlikely to gain benefits from using novaluron.

## 6. Evaluating IPM

As mentioned previously, Midsouth entomologists realized that insecticides would not solve the tarnished plant bug issue, but that an IPM approach was needed. They suggested adjusting where cotton was planted in relation to other crop hosts, and recognized the impact of cotton variety, planting date, fertility, and irrigation on tarnished plant bug populations in cotton [24]. For example, although *Lygus* spp. prefer to oviposit on cotton varieties with more leaf trichomes [60], varieties that do not have trichomes incur more square loss from tarnished plant bug [61]. Therefore, it is important that growers plant varieties with trichomes to minimize losses from tarnished plant bug. In Mississippi, later planting dates of cotton have more tarnished plant bugs and impact yield more than earlier planting dates [62]. Additionally, nitrogen application timing can influence tarnished plant bug numbers in cotton [63]. Finally, early irrigation during squaring can increase tarnished plant bug numbers, compared to later irrigation timings [64].

Using this knowledge, various systems approaches were trialed in North Carolina and Virginia. These were tested in various combinations, including applying an insecticide using the extension-recommended threshold, a calendar-based application schedule, and an arbitrarily high economic threshold, rotating insecticide classes, decreasing intervals between insecticide application, incorporating novaluron into the insecticide rotation, and a reduced rate of nitrogen in both a trichome-free variety and a variety with trichomes. This influence of landscape was known from previous studies and very little cotton is irrigated in the upper Southeast. Therefore, these factors were not tested. In these studies, as different IPM practices were added into the system, yield was preserved and net returns were not decreased [65]. This study demonstrated that IPM could deliver comparable net returns to growers as a non-IPM system while offering the additional benefits of IPM, such as conserving natural enemies, reducing insecticide usage, and delaying resistance development through insecticide rotation.

## 7. Incorporating a New Tool into the IPM System

ThryvOn cotton (Bayer Crop Science) expresses the Bt toxin Mpp51Aa2.834_16 [66,67] and, in the U.S. cotton production system, it has activity on *Lygus* spp. (primarily *Lygus hesperus* Knight and tarnished plant bug), *Pseudatomoscelis seriatus* Reuter [68,69], thrips, and *Franklinella* spp. [66,70]. ThryvOn causes mortality of *Lygus* spp. neonates and can reduce the mass of surviving nymphs [71].

ThryvOn cotton does not eliminate the entire tarnished plant bug population and may have variable impacts on their population levels. For example, in field studies with ThryvOn cotton, tarnished plant bug nymph numbers were 25–50% lower when compared to non-ThryvOn cotton [72,73]. However, in another study, tarnished plant bug nymph numbers on ThryvOn squares, flowers, and bolls were statistically the same as non-ThryvOn cotton [74]. Another study found a 44% reduction in the number of large-sized nymphs, a 40–42% reduction in the number of medium-sized nymphs, and a 33% reduction in the number of small-sized nymphs of a ThryvOn variety compared to a non-ThryvOn variety [75]. Finally, another study found a 48% reduction in the number of total nymphs (average of four locations) and a 60% reduction in the number of total nymphs (single location) of a ThryvOn variety compared to a non-ThryvOn variety [73]. These differences illustrate that the mortality of tarnished plant bug nymphs varies among populations or across environmental conditions.

Because of this variation and the lack of complete mortality, foliar insecticides are still recommended to manage tarnished plant bug in ThryvOn cotton (e.g., [76]). Depending on the situation, this may, or may not, result in a reduction in foliar insecticide applications. For example, in one study, ThryvOn varieties reduced the total number of insecticide applications for tarnished plant bug from 11 to 8 [72]. In contrast, in another study, the same number of average applications were needed for a ThryvOn and non-ThryvOn variety [75]. Furthermore, while current economic thresholds are recommended for ThryvOn varieties, preliminary studies indicate that thresholds may need to be adjusted to be specific to these varieties (B. Thrash personal communication). Additional studies will be needed to incorporate ThryvOn varieties into the current IPM system.

## 8. Conclusions

This tarnished plant bug IPM review in North Carolina represents a collaborative effort between the Midsouth and Southeastern entomologists. Many of the IPM strategies and tactics developed in the Midsouth were an excellent fit in North Carolina, including sampling methods, the economic threshold, the recommended sequence of insecticide classes across the season, and many of the same cultural tactics. However, more basic understanding of the pest’s life history and dispersal patterns in different geographies would help further refine recommendations for specific grower stakeholders. For example, the sequence of recommended insecticide classes across the season varies slightly between the regions, with modifications for number of expected applications and the use of novaluron later in the season in North Carolina. These results highlight the important modifications to tarnished plant bug pest management in cotton required regionally specific experimental evidence to tailor IPM practices for stakeholders. As new tactics are introduced into the system (e.g., ThryvOn cotton), additional experiments will be needed to maintain IPM efficacy.

## Figures and Tables

**Figure 1 insects-16-00164-f001:**
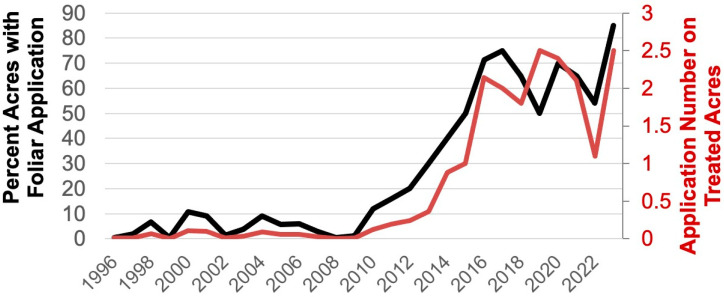
Percent of North Carolina cotton acres from 1996 to 2023 treated with a foliar insecticide application for tarnished plant bug (left *y*-axis, black line) and the average number of foliar insecticide applications for tarnished plant bug on treated acres (right *y*-axis, red line).

**Figure 2 insects-16-00164-f002:**
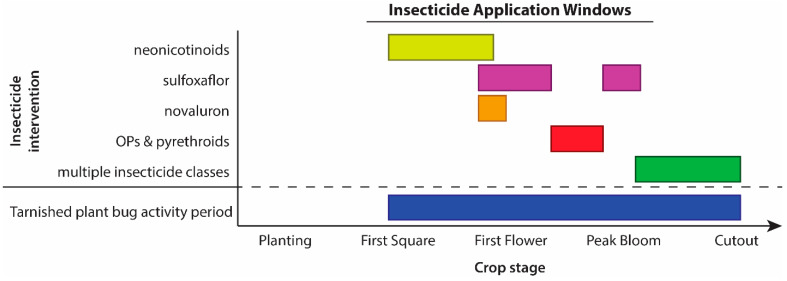
Example of a recommended insecticide application sequence from a Midsouth state. Yellow represents neonicotinoids, purple represents sulfoxaflor, orange represents novaluron, red represents organophosphates and pyrethroids, green represents multiple insecticide classes that can be tank mixed, blue represents the period during which tarnished plant bug is active. Adapted from Gore, Catchot, Cook and Dodds [23].

**Figure 3 insects-16-00164-f003:**
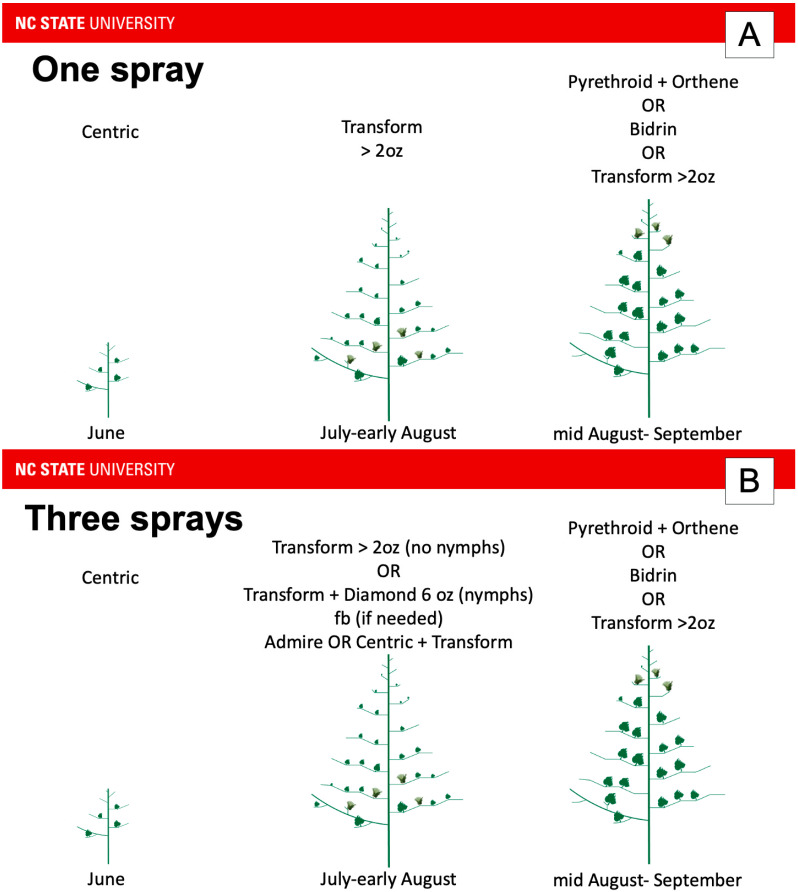
Recommended insecticide application sequence from North Carolina for (**A**) one expected application targeting tarnished plant bug, (**B**) three applications, and (**C**) four or more applications. Admire = imidacloprid; Centric = thiamethoxam (neonicotinoid class); Transform = sulfoxaflor; Diamond = novaluron; Orthene = acephate (organophosphate class); Bidrin = dicrotophos (organophosphate). Because tarnished plant bug pressure is not consistent across the state or from year-to-year, growers in an expected one-spray may have to apply an insecticide early, but not late in one year, or late, but not early in another year. However, their insecticide choice is predicated on the time of year that tarnished plant bug exceeds the economic threshold. For example, if the economic threshold was exceeded in June, the grower would be advised to apply Centric. However, if the economic threshold was exceeded in July, the grower would be advised to apply Transform alone if they expected to apply a single insecticide for tarnished plant bug during the year (**A**). This contrasts with the scenario where the grower expects to exceed the economic threshold multiple times a season (**B**). In this scenario, if the economic threshold is exceeded in July, the grower would be advised to apply Transform alone if nymphs are not present. If nymphs are present, the grower would be advised to apply Transform with Diamond. If a follow up spray is needed in July or early August, the grower would be advised to apply Transform with either Admire or Centric. If more applications are needed in July or early August, the grower would be advised to rotate applications of Transform and Diamond with applications of Transform and either Admire or Centric (**C**).

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
