# Peer review of "Establishing an IPM System for Tarnished Plant Bug (Hemiptera: Miridae) in North Carolina"

_insects, 2025, doi:10.3390/insects16020164_

Round 1
Reviewer 1 Report
Comments and Suggestions for Authors
Overall, this paper is a well-written review of the IPM system for managing tarnished plant bug in North Carolina and how the procedures compare with those established in the Mid-Southern U.S. Valuable information that should be published.
I have only noted a few grammatical corrections when reading through the manuscript:
Line 25: insert “in” between “pest” and “other” to read “persistent pest in other…”
Line 32: delete “to” at the end of the line
Line 51: delete “the U.S., with” – looks like this was left over from a deleted sentence.
Lines 54-55: delete “the most input costs incurred in”
Line 219: delete “the” in front of “North Carolina”
Line 257: replace “The” with “This” at the beginning of the sentence
Line 276: replace “and insecticide” with “an insecticide”
Line 289: I think you need to reference the ThryvOn technology (Bayer Crop Science); Also, provide some more information about the Mppp51As2.834_16 expressed in plants. Is it a Bt toxin?
Author Response
Overall, this paper is a well-written review of the IPM system for managing tarnished plant bug in North Carolina and how the procedures compare with those established in the Mid-Southern U.S. Valuable information that should be published.
I have only noted a few grammatical corrections when reading through the manuscript:
Line 25: insert “in” between “pest” and “other” to read “persistent pest in other…”
- Line 20 change made
Line 32: delete “to” at the end of the line
- Line 27 change made
Line 51: delete “the U.S., with” – looks like this was left over from a deleted sentence.
- We deleted it
Lines 54-55: delete “the most input costs incurred in”
- We deleted it
Line 219: delete “the” in front of “North Carolina”
- We deleted it
Line 257: replace “The” with “This” at the beginning of the sentence
- Change made
Line 276: replace “and insecticide” with “an insecticide”
- Change made
Line 289: I think you need to reference the ThryvOn technology (Bayer Crop Science); Also, provide some more information about the Mppp51As2.834_16 expressed in plants. Is it a Bt toxin?
- We have added a reference of ThryvOn cotton (Bayer Crop Science) and noted that it is a Bt toxin
Reviewer 2 Report
Comments and Suggestions for Authors
Reviewer comments insects-3412569
Line 51. Some words are missing in the sentence. Please rephrase it.
„seed. the U.S., with These effects can delay maturity and lead to yield losses [1-5].”
In line 56 you say: “The stink bug complex, another group of piercing-sucking pests, grew in prominence during the same period.”
I wouldn’t say it is another group of piercing-sucking pests, it is the same group actually, but a different true bug’s family, with just a bit different behavior. Therefore, I suggest rephrasing this sentence in a manner that those are similar insects.
Line 60. Here you mention another pest insect species, Anthonomus grandis Boheman, which is different order and family, so put it in the brackets (Coleoptera, Curculionidae). This is another group of pest insects with chewing mouth parts, and different biology, why it should be emphasized.
Line 78. You mention another insect pest, Helicoverpa zea Boddie, different from already mentioned so I suggest adding order and family in the brackets
Lines 175, 176 I suggest the same as in line 78, name order and family of tobacco budworm, cause it is different from tarnished or stink bugs.
Lines 185-188 Please add reference/s for these statements: During 2016, independent crop consultants in northeastern North Carolina began reporting poor control using pyrethroids for tarnished plant bug. A replicated small-plot trial in that region demonstrated that total plant bug numbers were statistically the same in plots treated with a pyrethroid compared to untreated plots.
Lines 189-190 However, tarnished plant bug numbers were well-controlled in plots where the pyrethroid had or has been combined with an organophosphate.
Line 289. You should explain what is ThryvOn cotton expresses Mpp51Aa2.834_16, for readers unfamiliar with it.
Line 290 you mention “..it has activity on Lygus spp. “, this must be better explained in which way ThryvOn cotton has activity on Lygus: what exact activity? Either in killing bugs, repellent, or something else.. you must be precise
Lines 299-301 “Another study found a 44% reduction of large-sized nymphs, a 40-42% reduction of medium-sized nymphs, and a 33% reduction of small-sized nymphs of a ThryvOn variety compared to a non-Thryvon variety [75].” Is the reduction related to the number of medium-sized nymphs or the size/development/biology… you must be more precise in explaining this.
Line 302 Explain better “..48% reduction of total nymphs..”. The total number of nymphs, nymph population, etc.?! Be precise in the whole paragraph (lines 294-305 should be rephrased in a way for better understanding)
Lines 321-322 You can’t conclude that the results “…highlight the generality of tarnished plant bug life history and interaction with the focal crop.” cause you didn’t take into consideration the life history of bugs, only their number based on monitoring. You considered the population's development in time, and their spreading, which is not the same.
Author Response
Line 51. Some words are missing in the sentence. Please rephrase it.
„seed. the U.S., with These effects can delay maturity and lead to yield losses [1-5].”
- Looks like this is in line 44 and not line 51. We have fixed the sentence
In line 56 you say: “The stink bug complex, another group of piercing-sucking pests, grew in prominence during the same period.”
I wouldn’t say it is another group of piercing-sucking pests, it is the same group actually, but a different true bug’s family, with just a bit different behavior. Therefore, I suggest rephrasing this sentence in a manner that those are similar insects.
- Excellent point. We now state that stink bugs are also piercing-sucking pests
Line 60. Here you mention another pest insect species, Anthonomus grandis Boheman, which is different order and family, so put it in the brackets (Coleoptera, Curculionidae). This is another group of pest insects with chewing mouth parts, and different biology, why it should be emphasized.
- Added order and family as suggested
Line 78. You mention another insect pest, Helicoverpa zea Boddie, different from already mentioned so I suggest adding order and family in the brackets
- Added order and family as suggested
Lines 175, 176 I suggest the same as in line 78, name order and family of tobacco budworm, cause it is different from tarnished or stink bugs.
- Added order and family as suggested (lines 155-156)
Lines 185-188 Please add reference/s for these statements: During 2016, independent crop consultants in northeastern North Carolina began reporting poor control using pyrethroids for tarnished plant bug. A replicated small-plot trial in that region demonstrated that total plant bug numbers were statistically the same in plots treated with a pyrethroid compared to untreated plots.
- Reference for this was already provided after the next sentence. We moved the reference forward for this suggestion
Lines 189-190 However, tarnished plant bug numbers were well-controlled in plots where the pyrethroid had or has been combined with an organophosphate.
- Had is correct
Line 289. You should explain what is ThryvOn cotton expresses Mpp51Aa2.834_16, for readers unfamiliar with it.
- We have added a reference of ThryvOn cotton (Bayer Crop Science) and noted that it is a Bt toxin
Line 290 you mention “..it has activity on Lygus spp. “, this must be better explained in which way ThryvOn cotton has activity on Lygus: what exact activity? Either in killing bugs, repellent, or something else.. you must be precise
- We explain this in the next sentence “ThryvOn causes mortality of Lygus neonates and can reduce the mass of surviving nymphs”
Lines 299-301 “Another study found a 44% reduction of large-sized nymphs, a 40-42% reduction of medium-sized nymphs, and a 33% reduction of small-sized nymphs of a ThryvOn variety compared to a non-Thryvon variety [75].” Is the reduction related to the number of medium-sized nymphs or the size/development/biology… you must be more precise in explaining this.
- We have added “in the number” before these statements to be more precise
Line 302 Explain better “..48% reduction of total nymphs..”. The total number of nymphs, nymph population, etc.?! Be precise in the whole paragraph (lines 294-305 should be rephrased in a way for better understanding)
- We have added “in the number” before these statements to be more precise
Lines 321-322 You can’t conclude that the results “…highlight the generality of tarnished plant bug life history and interaction with the focal crop.” cause you didn’t take into consideration the life history of bugs, only their number based on monitoring. You considered the population's development in time, and their spreading, which is not the same.
- We modified the sentences to read “These results highlight the important modifications to tarnished plant bug pest management in cotton required regionally specific experimental evidence. However, more basic understanding of the pest's life history and dispersal patterns in different geographies would help further refine recommendations for specific grower stakeholders.”
Reviewer 3 Report
Comments and Suggestions for Authors
This review outlines the considerations and information required to modify an IPM plan for tarnished plant bug from one cotton growing region to implement in another. Overall, the information presented was thorough and the differences among regions was highlighted. I had only a few comments, mainly dealing with clarifying statements or concepts. Even though this is a review article, certain terminology and aspects of the pest system need to be better clarified for better understanding by the general reader. I feel that if my comments are addressed this paper would be acceptable for publication.

Author Response
This review outlines the considerations and information required to modify an IPM plan for tarnished plant bug from one cotton growing region to implement in another. Overall, the information presented was thorough and the differences among regions was highlighted. I had only a few comments, mainly dealing with clarifying statements or concepts. Even though this is a review article, certain terminology and aspects of the pest system need to be better clarified for better understanding by the general reader. I feel that if my comments are addressed this paper would be acceptable for publication.
In
- Change made
Remove sentence fragments and reword.
- Change made
This region needs to be defined.
- We now list the states and the region within the states
Reword sentence for clarity.
- We reworded the sentence for clarity
Is this the same region as the midsouth mentioned above?
- No and we have now defined it for clarity
I suggest using 'review' instead of 'case study' or 'study'.
- We have changed the two mentions of case study to review
Is there a document or website with the midsouth IPM system/plan/recommendations published, or have IPM practices developed in each state from information gained in research conducted in midsouth states? As written it seems like there is a consolidated plan that growers can reference, but if there is it is not highlighted (and the most likely references I found were the thresholds for TPB).
- Excellent point. We have now referenced several online publications with broad recommendations applicable to Midsouth growers
A couple of suggestions. First, as suggested in the first paragraph, the regions (midsouth vs southeast) need to be defined, as they may mean different things to different readers. Second, I think a conclusion sentence is needed for this paragraph detailing what you are about to present in the sections that follow. The language of this paragraph alludes to a NC IPM plan and how it is modified from a midsouth plan. However, stating what is needed to develop an IPM plan and why the specific sections are covered would help orient the reader.
- We have now defined the Midsouth and Southeast cotton production areas. We have added two concluding sentences as suggested
Cultivated
- Change made
Strike field
- Change made
Was more
- Change made
Stike through comma
- Change made
Are either or both of these sampling recommendations to be conducted at random locations in the field, at field margins or X meters from the edge of a field? Previous sentences make it sound like targeting field margins for sampling is best, but the recommendation doesn't state where in the field to target.
- We clarified this as “throughout the field”
In a field
- We clarified this as “throughout the field”
Inaccurate sampling results
- Change made
Could this heading and heading 3.2 and 3.3 be main headings instead (3.1=>4, 3.2=>5, _3.3=>6, current 4=>7, and current 6=>8)? I can see having 'Thresholds' being a subsection of 'Sampling', but 'Insecticides' and 'Evaluating IPM' don't seem to fit as well under 'Sampling'. I suggest making each of these subsections a main section instead.
- Yes, this was simply a formatting error. We have corrected this in our submission
Is this allowed by the journal? Check the style guide.
- We did not see that it was prohibited
This wording is unclear (it sounds like the presence of bollworm caused Bt cotton not to grow, but I think you are referring to pest outbreaks despite having planted Bt varieties that should control bollworm). Rephrase for clarity.
- Rephrased for clarity
Again, this wording seems incomplete. Rephrase for clarity.
- Rephrased for clarity
Highlighted bollworm
- This should make sense in the new context now
A couple of comments: First, this figure is not referenced in the text. Perhaps reference it in one of the two following paragarphs? Second, a suggestion for editor/publisher: resize figure, so that all three panels (A, B, and C) are on the same page.
- We have now referenced the figure in the text
This figure is not as intuitive as it should be. For example, how is A representing a one-spray scenario when there are three compounds applied throughout the year? Is just one intended for TPB? Which one? The same lack of clarity applies to B and C. I suggest expanding the explanation of each scenario (i.e., number of compounds applied at each developmental interval of the plant) in the figure title, labeling the applications/ compounds more clearly, or both.
- We have modified the legend to clarify the figure
This compound is not mentioned in Fig. 3. Should it be (i.e., which application should be mixed with novaluron)?
- We have now mentioned this in the legend
Protein? Gene? Toxin? Please provide more context.
- Now specified as Bt toxin
List this one following tarnished plant bug and before thrips, since it is also a Mirid.
- Change made